# Modulation of Morphology and Glycan Composition of Mucins in Farmed Guinea Fowl (*Numida meleagris*) Intestine by the Multi-Strain Probiotic Slab51^®^

**DOI:** 10.3390/ani11020495

**Published:** 2021-02-13

**Authors:** Salvatore Desantis, Livio Galosi, Nicoletta Santamaria, Alessandra Roncarati, Lucia Biagini, Giacomo Rossi

**Affiliations:** 1Department of Emergency and Organ Transplantation (DETO), University of Bari Aldo Moro, S.P. 62 per Casamassima Km 3, 70010 Valenzano (Bari), Italy; nicoletta.santamaria@uniba.it; 2School of Biosciences and Veterinary medicine, University of Camerino, Via Circonvallazione 93/95, 62024 Matelica (M.C.), Italy; alessandra.roncarati@unicam.it (A.R.); lucia.biagini@unicam.it (L.B.); giacomo.rossi@unicam.it (G.R.)

**Keywords:** glycohistochemistry, lectin, sialic acid, fucose, probiotics, gut

## Abstract

**Simple Summary:**

In the poultry industry, several studies demonstrate the positive effects of probiotic administration on induction of intestinal gene expression, physiology, immunology, morphology and mucus composition. The mucus layer covers the epithelium of the gastrointestinal mucosa, protects it against physical and chemical injuries caused by food, microbes adhesion, and microbial metabolites, promotes the gut content elimination and modulates water and electrolyte absorption. Mucins are glycoproteins that play a key role in constituting the intestinal mucus layer, and their production takes place in the goblet cells. This study examined the effects of the multi-strain probiotic Slab51^®^ on the morphology and carbohydrate composition of intestinal glycoproteins of 40 guineafowl (*Numida meleagris*), averagely weighing 110 ± 0.99 g, during a grow-out cycle that lasted 120 days. Samples from different anatomical tracts of intestine, collected after slaughtering, were processed for morphological, morphometric, conventional and lectin glycohistochemical studies. Compared with control samples, probiotic group revealed significant increase in morphological parameters and goblet cells expression per villus in all investigated tracts as well as region-specific changes in carbohydrate composition of glycoproteins of the mucus layer.

**Abstract:**

Probiotics have become highly recognized as supplements for poultry.Since gut health can be considered synonymous withanimal health, the effects of probiotic Slab51^®^ on the morphology and the glycan composition of guineafowlintestine were examined. The probiotics were added in drinking water (2 × 10^11^ UFC/L) throughout the grow-out cycle.Birds were individually weighed andslaughtered after four months. Samples from the duodenum, ileum and caecum were collected and processed for morphological, morphometric, conventional and lectin glycohistochemical studies.The results were analyzed for statistical significance by Student’s *t* test. Compared with control samples, probiotic group revealed (1) significant increase in villus height (*p* < 0.001 in duodenum and ileum; *p* < 0.05 in caecum), crypt depth (*p* < 0.001 in duodenum and caecum; *p* < 0.05 in ileum) and goblet cells (GCs) per villus (*p* < 0.001) in all investigated tracts; (2) increase in galactoseβl,3N-acetylgalacyosamine(Galβl,3GalNAc)terminating O-glycans and αl,2-fucosylated glycans secretory GCs in the duodenum; (3) increase in α2,6-sialoglycans and high-mannose N-linked glycans secretory GCs but reduction in GCs-secreting sulfoglycans in the ileum; (4) increase in Galβl,3GalNAc and high-mannose N-linked glycans secretory GCs and decrease in GCs-producing sulfomucins in the caecum; (5) increase in the numbers of crypt cells containing sulfate and non-sulfated acidic glycans. Overall, dietary Slab51^®^ induces morphological and region-specific changes in glycoprotein composition of guinea fowl intestine, promoting gut health.

## 1. Introduction

The gastro-intestinal mucosal surface of vertebrates is covered with mucus, a viscous gel-like layer which is produced by goblet cells. This mucus layer is the single large physical barrier which prevents invasion of the intestinal epithelial cells by gut pathogen bacteria and viruses, and it may selectively facilitate adherent growth of normal resident gut microbiota [1]. Intestinal mucus consists mainly of secreted mucin glycoproteins. The carbohydrate side chains of the protein backbone of intestinal mucins are composed of fucose (Fuc), N-acetylgalactosamine (GalNAc), N-acetylglucosamine (GlcNAc), galactose (Gal), glucose (Glc), and mannose (Man), and N-acetilneuraminic acid (NeuNAc, sialic acid) and their ester sulfates [2,3].

The common classes of glycans are N- or O-glycans. An N-glycan is a sugar chain covalently linked to an asparagine residue of a polypeptide chain. An O-glycan is frequently linked to a serine or threonine residue of the polypeptide [4]. O-glycosylated mucin contributes up to 80% of the molecular weight of mucins [5]. The high level of glycosylation enables mucins to function as a protective barrier by lubricating the epithelium and preventing degradation of the protein backbone by proteases [6,7]. Moreover, mucin glycans can modulate cell adhesion, serve as ligands for cell surface receptors, take part in host–pathogen interactions and serve as energy sources for both commensals and pathogens [1]. Lastly, defects in mucin glycosylation can lead to severe inflammation and susceptibility to infection, and the glycans themselves have been shown to be ligands that can block the binding of microorganisms [8,9]. In birds, the host defense against infection can result in differences in mucin expression, as is the case of necrotic enteritis infection in the mucosa of the small intestine [10].

Prevention and control of poultry diseases should be necessary in order to avoid large economic losses. Public concern about microbial antibiotic resistance has led to a ban on sub-therapeutic antibiotic usage as a growth promoter for poultry and other livestock in Europe, and there are potential bans in other countries. Probiotics are one of several approaches that have been considered as alternatives to antibiotic growth promoters in poultry production [11]. Probiotics are defined as “live microorganisms that, when administered in adequate amounts, confer a health benefit on the host” [12]. The probiotic microbes contribute in protection of the host against intestinal pathogens through colonization of the intestine [13] and stimulating immune responses by promoting the endogenous host defense mechanisms and modulating the mucosal immune system [11,14].

Gut health can be considered a synonymous to animal health and is of vital importance to animal performance [15]. Gut health depends on the maintenance of the delicate balance between the host, intestinal microbiota, intestinal morphology, immunesystem and dietary compounds. There is evidence that gut microbiota may modulate synthesis and composition of mucins, which constitute a digestion- and absorption-assisting medium and represent the first line of defense for intestinal epithelium against foreign bacteria and other pathogens [1,16,17].

Recently, it has been demonstrated that the supplementation of a commercial multistrain probiotic (Slab51^®^*,* administered in drinking water, 2 × 10^11^ UFC/L) positively affects the morphology and microbiota diversity of guineafowl (*Numida meleagris*) intestine [18], and this species has been indicated as a suitable alternative to laying hens due to their higher resistance to diseases, ability of adaptation to different environmental conditions and in terms of eggshell quality and therefore egg safety [19]. Considering that Slab51^®^ induces changes in the carbohydrate composition of glycans secreted along intestinal tracts of mammals [20,21], we aimed to evaluate the effect of the aforementioned probiotics on the glycan composition of glycoproteins produced in the intestine, with particular attention to the goblet cells mucins, of guineafowl. The glycan characterization was carried out by the use of both conventional and lectin histochemistry.

## 2. Materials and Methods

### 2.1. Animals

This study was performed during a normal zootechnical cycle, avoiding any animal suffering, and no samples were collected from live animals, according to the Italian Legislative Decree 26/2014.

At 10 days of age, 40 healthy unsexed guinea fowls (*Numida meleagris*), averagely weighing 110 ± 0.99 g, were randomly assigned to two groups: control group (C) received water without any additive, while treated group (P) received drinking water supplemented with Slab51^®^ (trademarked as SivoMixx^®^, Ormendes SA, Jouxtens-Mezery, Switzerland) at the dosage of 2 × 10^11^ UFC/L. Slab51^®^ is a commercial multi-strain probiotic containing 200 billion lactic acid bacteria per 1.5 g of product, comprised of the following strains: *Streptococcus thermophilus* DSM 32245/CNCM I-5570, *Bifidobacterium lactis* DSM 32246/CNCM I-5571, *Bifidobacterium lactis* DSM 32247/CNCM I-5572, *Lactobacillus acidophilus* DSM 32241/CNCM I-5567, *Lactobacillus helveticus* DSM 32242/CNCM I-5573, *Lactobacillus paracasei* DSM 32243/CNCM I-5568, *Lactobacillus plantarum* DSM 32244/CNCM I-5569 and *Lactobacillus brevis* DSM 27961/CNCM I-5566. Both groups were housed in two adjacent sheds (12 m^2^ each), equipped with 15 cm fresh straw litter, under controlled photoperiod (14 h light, 10 h dark) and natural ventilation by means of small windows. Throughout the trial, both groups received ad libitum the same commercial pellet feed (Cruciani, Montappone, MC, Italy) as starter, followed by agrowing feed that changed in proximate composition in relation to the age of the animals (Table 1). At the end of the cycle, all the birds of both groups were weighted with an electronic balance (mod. ACS-A9, My Scale, Foggia, Italy), and body weight was recorded. Animals were slaughtered by electrical stunning and bleeding at 120 days of age.

### 2.2. Sampling and Tissue Preparation

Segments of approximately 3 cm were collected from the duodenum loop, the last tract of the ileum and the right caecum and fixed in 4% (*v*/*v*) phosphate-buffered saline paraformaldehyde for 24 h at 4 °C. The samples were then dehydrated through a graded series of ethanol and embedded in paraffin wax. Serial sections (5 μm thick) were cut, and after being de-waxed with xylene and hydrated in an ethanol series of descending concentrations, they were stained with hematoxylin-eosin for morphological studies and by conventional histochemical procedures or lectin histochemistry for mucin characterization.

### 2.3. Conventional Histochemistry

Sections were treated with (1) the periodic acid-Schiff (PAS) reaction for neutral glycans [22]; (2) Alcian Blue pH 2.5 (AB2.5) for sulphate esters and carboxyl groups in mucins [23]; (3) combined high iron diamine/AB2.5 (HID/AB2.5) for simultaneous staining of sulfted and non-sulfated acidic glycans [24]; and (4) an AB2.5/PAS sequence to reveal combinations of acidic and neutral mucins.

### 2.4. Lectin Histochemistry

To characterize the mucin composition, tissue sections were rinsed in 0.05 M Tris-HCl-buffered saline (TBS), pH 7.4, and incubate at room temperature (RT) for 1 h in the dark with appropriate dilutions of five fluorescent lectins (Table 2) diluted in the TBS. All lectins were obtained from Vector Laboratories (Burlingame, CA, USA) except for Maackia amurensis lectin II (MAL II) that was purchased by Glycomatrix (Dublin, OH, USA). After three rinses in TBS, slides were mounted in Fluoroshield with DAPI (Sigma-Aldrich, St. Louis, MI, US). Each experiment was repeated twice for each sample. Controls for lectin staining included (1) substitution of the substrate medium with buffer without lectin and (2) incubation with each lectin in the presence of its hapten sugar. All control experiments gave negative results.

### 2.5. Sialidase Treatment

Sialic acid residues were directly detected with MAL II and SNA and indirectly demonstrated with identification of its subterminal GalNAc residue by the binding with PNA lectin (specific for Galβ1, 3GalNAc), without and with prior sialidase (s) digestion. Before incubation with the above cited lectins, sections were incubated at 37 °C or 16 h in 0.86 U/mg protein of sialidase (neuraminidase) (Type V, from *Clostridium perfringens*) dissolved in 0.1 M sodium acetate buffer, pH 5.5, containing 10 mM CaCl_2_. Prior to the neuraminidase treatment, a saponification technique was performed to render the enzyme digestion effective, with 0.5% KOH in 70% ethanol for 15 min at RT [25]. As controls of the enzyme digestion procedure, some sections were incubated in the enzyme-free buffer solution under conditions of the same duration and temperature. In control sections, cleavage of sialic acid was not evident. Slides were observed with the light photomicroscope Eclipse Ni-U (Nikon, Tokyo, Japan) at 20× magnification and photographed with a digital camera (DS-U3, Nikon, Japan). The images were analyzed by the image-analyzing program NIS Elements BR (Version 4.20) (Nikon, Tokyo, Japan).

### 2.6. Morphometry and Statistical Analysis

Hemoxylin-eosin-stained sections of 10 well-oriented villi and crypts of duodenum, ileum and caecum from each animal were photographed with a 10× lens using a light microscope (Eclipse Ni-U; Nikon, Tokyo, Japan) and used to measure the villus height (VH) and the crypt depth (CD). The total number of goblet cells per villus, the number of goblet cells with different types of mucins as distinguished by conventional and lectin histochemistry were determined by counting of 10 well-oriented duodenum, ileum and caecum sections. The fluorescence signal intensity of stained GCs and no-stained epithelial cells (selected as background) were measured with the image-analyzing program NIS Elements BR (Version 4.30) (Nikon, Tokyo, Japan). The fluorescence intensity of GCs was calculated after subtraction of the background signal, and the mean intensity value was computed.

Values were expressed as means ± standard error (SE). The results were evaluated for statistical significance by Student’s *t* test.

## 3. Results

### 3.1. Animals

At the end of the trial, P showed a favorable final mean body weight (1820.45 ± 209 g) without significant differences in comparison with control group (1754.05 ± 140 g).

### 3.2. Morphometry

The examined organs from C and P samples did not show either macroscopic or histological lesions. The intestinal mucosa consisted of villi and basal crypts. The mucosa was covered by a simple columnar epithelium, and GCs were scattered among the columnar cells. The morphometric evaluations revealed that the probiotic blend significantly increased the villus height and the crypt depth compared with the control group (Table 3).

### 3.3. Histochemistry

AB 2.5/PAS sequential staining showed that the dietary probiotic increased the number of goblet GCs per villus throughout the entire intestine compared with the C (Table 3). AB 2.5/PAS sequential staining showed that most GCs of the intestinal villi from both Cand P produced only acidic mucins (AB2.5 positivity) (Figure 1 and Figure 2), whereas few GCs produced either only neutral mucins (PAS positivity) or a mixture of neutral and acidic mucins (staining with both AB2.5/PAS) (Figure 1; top inserts of Figure 2A–F).

In C, the difference between the percentage of PAS-positive GCs and AB2.5/PAS-positive GCs was not statistically significant in the duodenum and the caecum, whereas PAS-positive GCs were significantly lower (*p* = 0.034) than AB2.5/PAS GCs in the ileum (Figure 2A,C,E). In P, the percentage of PAS-positive and AB2.5/PAS-positive GCs was not statistically significant in the duodenum, whereas a significant increase in the AB2.5/PAS-positive GCs percentage was detected in the ileum (*p* = 0.031) and in the caecum (*p* = 0.023) (Figure 2B,D,F). The comparison between C and P did not reveal a significant difference in the duodenum, whereas it displayed a significant increase in the AB2.5/PAS-positive GCs in the ileum (*p* = 0.033) and in the caecum (*p* = 0.046). The caecum also showed a significant (*p* = 0.036) increase in the GCs secreting only acidic mucins (AB 2.5 positivity). AB 2.5/PAS sequential method stained weakly the epithelial cells and strongly the apical surface of the villi with PAS reaction (top insets of Figure 2). Moreover, the crypt epithelial cells displayed strong positivity for AB 2.5 in all the investigated intestinal tracts with a larger presence of positive cells in P (bottom insets of Figure 2).

The combination HID/AB2.5 method showed a strong predominance of HID-positive GCs over AB 2.5-positive GCs in all the intestinal tracts of both C and P (Figure 3 and Figure 4). In C, all duodenal GCs were HID-positive (Figure 3 and Figure 4A), whereas the ileum and the caecum showed a very low presence of AB2.5- and HID/AB2.5-positive GCs (Figure 3 and Figure 4C,E). Statistical analysis did not reveal significant difference between the two latter GCs subtypes in both the ileum and caecum. In P, the appearance of AB 2.5- and HID/AB2.5-positive GCs was observed in all investigated intestinal tracts (Figure 3). The percentage difference between AB2.5- and HID/AB 2.5-positive GCs was not statistically significant in the duodenum and caecum, whereas HID/AB 2.5-positive GCs significantly (*p* = 0.041) predominated over AB2.5-positive GCs in the ileum.

The comparison of GCs percentage from C and P guinea fowls revealed the following: (1) a significant reduction in HID-positive GCs in the duodenum (*p* = 0.014), the ileum (*p* = 0.001) and the caecum (*p* = 0.021); (2) the appearance of AB2.5- and HID/AB2.5-positive GCs in the duodenum; (3) a significant increase (*p* = 0.006) in HID/AB2.5-positive GCs in the ileum and a significant (*p* = 0.015) increase in AB2.5-positive GCs in the caecum.

HID/AB2.5 method did not stain the epithelium and the apical surface of the villi. As for the crypts, apical HID positivity was found in cells of ileum and caecum of C and throughout the intestine of the P birds, which showed a greater number of positive cells when compared to C samples (see insets below Figure 4).

### 3.4. Lectin Histochemistry

Peanut agglutinin (PNA), binding mucin-type glycans terminating with Galβl, 3GalNAc, reacted with the GCs in all intestinal tracts (Figure 5).

Probiotic supplementation significantly increased the number of GCs in the duodenum (*p* < 0.001) and the caecum (*p* < 0.05) (Figure 5 and Figure 6).

Sialidase (s) pre-treatment, which is used to indirectly reveal sialic acid linked to Galβl, 3GalNAc, produced a significant (*p* < 0.001) increase in the PNA-positive GCs in the villi of all intestinal tracts of Cguineafowls and in the ileum and caecum of P animals (Figure 5 and Figure 6). The evaluation of PNA staining signals revealed a significant (*p* < 0.001) increase in duodenal and ileal GCs of P guinea fowls (Figure 7 and Figure 5C,G). Sialidase incubation significantly (*p* < 0.001) increased the PNA staining intensity of GCs in all intestinal tracts (Figure 7). PNA reactivity was not observed in the columnar epithelial cells and luminal surface of the villi (Figure 5), whereas PNA-binding sites were detected in the apical zone of the cell crypts (Figure 5O) of all investigated intestinal tracts without significant differences between C and P. Sialidase treatment did not affect the PNA staining pattern of villus and crypt epithelial cells.

*Sambucus nigra* lectin (SNA), specific for Neu5Acα2,6Gal/GalNAc, did not react with the GCs of duodenum, whereas it bound GCs of the ileum and the caecum of C fowls (Figure 6; Figure 8A,C,E). Probiotic supplementation induced the staining of some SNA-positive GCs in few duodenal villi (Figure 8B) and significantly (*p* < 0.05) increased the number of ileal GCs (Figure 6; Figure 8D,F). In addition, probiotic administration induced a significant increase in the signal intensity of SNA in the GCs of the ileum, whereas it significantly (*p* < 0.05) decreased the presence of α2,6-sialomucins in the GCs of the caecum (Figure 7). This lectin did not find binding sites in the epithelium lining the villi and in the crypts of all investigated intestinal tracts.

*Maackia amurensis* lectin-II (MAL II), specific for α2,3-sialoglycans, did not react with the GCs and the epithelium lining the villi and the crypts of all investigated intestinal samples, whereas it bound some cells of the lamina propria (Figure 8G).

Concanavalin A (Con A), specific for mannosylated glycans, bound GCs in the villi of all the intestinal tracts. These cells were hardly distinguishable from the positive epithelial cells in some slides (Figure 9).

In P guinea fowls the number of distinguishable Con A-positive GCs significantly (*p* < 0.05) increased in the ileum and the caecum (Figure 6), whereas the fluorescence intensity increased in the GCs of duodenum and decreased in the ileal and the caecal GCs (Figure 7; Figure 10). The epithelial cells and the luminal surface of the villi as well as the crypt cells (Figure 10O) displayed Con A-binding sites along the entire intestine, and their staining intensity was not affected by the administration of probiotics.

*Lotus tetragonolonus* agglutinin (LTA) bound αL-fucose terminating mucins in GCs of all the intestinal tracts of Cand P guinea fowls, showing an increase in the number of positive GCs moving from duodenum to ileum to caecum (Figure 6). P guinea fowls displayed a significant increase in the number (*p* < 0.001) and the fluorescence intensity (*p* < 0.05) of LTA-positive GCs in the duodenal tract, and reduction (*p* < 0.001) of the staining in ileal and caecal GCs (Figure 6, Figure 7 and Figure 11). The apical surface of the villi and the epithelial cells constituting the crypts (Figure 11M) reacted with LTA without significant difference between Cand P samples.

*Ulex europaeus* agglutinin (UEA I) evidenced the presence of αl,2-fucosylated glycans only in epithelial cells lining the villi of the intestinal tracts without difference between control and treated fowls.

## 4. Discussion

Probiotics have become highly recognized as supplements for humans and animals, including poultry, because of their beneficial effects on health and well-being. Since gut microbiota may regulate synthesis and composition of mucins, this study, utilizing histological methodologies, demonstrated that dietary administration of the multi-strain probiotics Slab51^®^ produces regional effects on the intestinal morphology and on the carbohydrate composition of glycan expressed along intestinal tracts of guinea fowl (*Numida meleagris)*. At the end of the trial, P group showed a favorable final mean body weight, which was slightly higher compared to C, although not statistically significant. Guinea fowls are ancestral birds. Currently, the bird bred for meat production is not morphologically selected with respect the wild conspecific specimens that live insub-Saharan Africa, and no appreciable morphological differences are shown between sexes, despite thatadult males are heavier than females [26,27]. Further studies, using a more representative and sexed population of guinea fowls, will be necessary to establish if also in this species the multi-strain probiotic Slab51^®^ can guarantee high productive performances as observed in other species [28].

Histomorphometric analysis confirmed that the used probiotics induces a general increase in the villus height and crypt depth [18]. Intestinal crypts consist of pluripotent stem cells and cells that differentiate into enterocytes and GCs during migration along the crypts.Therefore, the length of villi is related to proliferating rate in the crypts and differentiating rate of villus epithelial cells. Crypt lengthening may be mediated by a spectrum of local, immune and neurohumoral factors [29] as well as by intestinal commensal microbiota [30,31]. The increase in intestinal villus height is related to the enhancement of absorptive surface area and subsequent satisfactory digestive enzyme action and higher transport of nutrients, which improve growth performance [32].

Histochemical analysis using AB2.5-PAS revealed that probiotic supplementation induced a significant increase in the number of mucus-secreting GCs per villus in all the investigated intestinal tracts. This effect has an important physiological importance, since GCs influence the quantity and quality of the mucus layer that forms a separation between the lumen and the host [33]. The mucus layer serves to lubricate the mucosal surface, to protect underlying intestinal epithelial cells from chemical and mechanical stress and bacteria, and as a transport medium between luminal contents and epithelial cells [6,7]. Furthermore, it provides a habitat for commensal bacteria and signals to the underlying immune system [1,10].

Mucins perform a very important function in the modulation of the intestinal microbiota, as they contain a very large number of glycoproteic motifs of attack for bacteria, contributing to the mechanism of the non-immune exclusion of the intestinal microbiota [34]. Several factors such as hormones (neuropeptides), inflammatory mediators and microbial colonization can affect the activity of GCs and the secretion of mucin [1,35]. The observed increase in the GC due to supplementation of probiotics mixture can be attributed to their mucin gene regulation, acceleration of differentiation and immunostimulating effects of probiotics [36]. Although the function of the GCs is largely related to mucus secretion, GCs may play an important role in epithelial cell repair following damage to the GI mucosa [37].

Mucins are glycoproteins that can be classified into neutral and acidic types. When the latter terminate with sialic acids or sulphate groups, they constitute the sialomucins and the sulfmucins, respectively [17]. In the entire intestine of both C and P guinea fowls AB2.5-PAS staining demonstrated that the percentage of GCs producing acidic mucins markedly predominated over those synthesizing only neutral glycans or a mixture of both neutral and acidic mucins. A scarce presence of GCs containing neutral mucins has been reported in the intestinal villi of small intestine of Hubbard chicks [38] and Cobb 500 broiler chicks [39] but not in the small intestine of Cobb chicks [40,41] and in the entire intestine of Label Hubbard hybrid chickens [42]. Since GCs producing neutral mucins were mainly present in the duodenum, their secretion together with the secretion of bile and pancreatic juice could be involved in the neutralization of the acidic pH of gastric juices entering the duodenum [43]. Moreover, it has been reported that the production of neutral mucins could also serve as a protective mechanism against invasion by pathogenic bacteria [44,45]. In the current study, probiotic supplementation increased the number of GCs containing only neutral mucin in the ileum and GCs producing mixed (neutral and acidic) mucins in the ileum and in the caecum. During the maturation process cycle of a goblet cell, the mucin composition gradually changes from neutral to acidic [38].

Regarding the high presence of GCs producing acidic mucins, a strong prevalence of sulfated mucin secretory GCs (HID positivity) over GCs containing acidic carboxylated (sialylated) mucins (AB 2,5 positivity) in all intestinal tracts of both C and P birds was observed. Specifically, in C the GCs containing only acidic carboxylated mucins were not detected in the duodenum, whereas they did not exceed 3% in the other intestinal tracts, which also exhibited 3–4% of GCs containing a mixture of both sulfated and acidic carboxylated mucins. Some authors found that the most recurrent structural motif in chicken intestine was sulphation without differences between the small and large intestines [46]. This justifies the low abundance of GCs stained only with PAS and with AB 2.5. Sulfated glycans are in most cases extensions of the neutral or sialylated glycans. Except for the duodenum, the probiotic blend induced an increase in the percentage number of GCs secreting sialylated mucins or both sulfate and sialylated mucins. This could depend on the increased number of AB 2.5- and HID-positive cells in the crypts. Sulfated residuals are involved in regulating the interactions with microorganisms and parasitic helminths, as well as in preventing inflammatory disorders [47,48,49]. Sulfation together with sialylation increases the overall negative charge of mucins and can protect the host from mucin degradation by glycosidases [50,51,52], thereby impeding colonization by enteric pathogens and reducing the gastrointestinal infections [40,46]. This finding suggests that probiotics used in the current study could have beneficial effects on health of the guineafowls, closely related to the increase in mucin secretion in birds treated with probiotics. Studies carried out in chickens demonstrate that intestinal mucins represent an important source of carbohydrates and nutrition for many bacterial species that have a fundamental role in maintaining the intestinal integrity of birds [34]. In particular, *Lachnospiraceae* along with *Ruminococcaceae* represent typical butyrate- and SCFAs-producing bacteria families [53] which proliferate when a greater amount of intestinal mucus is produced. Although there are not many studies on the microbial flora of guinea fowl, studies carried out on chicken and other animal species, treated with the same probiotic blend used in this study or similar products, indicate a characteristic increase in these families, as well as a characteristic increase in other mucin-degrading and strong butyrate producer gut bacteria, as *Bacteroides* spp. [14,54]. Butyrate has various positive properties, since it represents an important nutrition source for the enterocytes, stimulating a strong gut mucin production [55], and improving tight-junction integrity [56] in the intestinal epithelial layer. Butyrate it is also involved in the cellular differentiation and proliferation within the intestinal mucosa [57], and it is capable of reducing the inflammatory response as an anti-inflammatory effector [58]. All these proprieties promote a good health status in chickens, as well as in guinea fowls, inducing also better productive parameters. Therefore, the assessment of mucin types in the current research is indicative of overall mature and healthy guts with a well-developed mucin secretory architecture.

A greater quantity of GCs and a greater production of mucus at the ileal and caecal level allow the guineafowls treated with probiotics to have greater protection in host–pathogen interactions. The initial interaction between the coccidian parasite *Eimeriatenella* and the host intestinal epithelium, for example, must occur across the mucus interface. A recent study demonstrates that purified native chicken intestinal mucin, particularly these belonging to ileal and cecal tracts, inhibited parasite invasion into epithelial cells in vitro [59]. Recent studies have suggested that chicken mucins may be responsible for preventing colonization by bacterial pathogens and that enrichment of Neu5Ac and sulfate residues may prevent mucus colonization by *Campylobacter jejuni* [46].

In the current study we used the lectin histochemistry to detect the presence and the distribution of sialogycans, O- and N-glycans and fucosylated glycoproteins. Lectins are particularly well suited for discriminating glycoconjugates because of their specificity and ability to distinguish sugar isomers, as well as branching, linkage and terminal modifications of complex glycans [60].

Sialic acid exhibits pronounced chemical diversity in structure and linkage so that the total diversity of sialoglycans constitutes the “sialome” [61]. In this study the presence of disialyl T sequence sialic acidα2,3Galβ1,3(±sialicacidα2,6)GalNAc, Neu5Acα2,6Gal/GalNAc, and sialic acid-β-Gal(1,3)GalNAcwas investigated by means ofMAL II, SNA and sialidase-PNA, respectively. α2,3-linked sialic acid was absent in villi and crypts of all investigated samples, whereas α2,6-sialoglycans were found in the GCs of the ileum and caecum from C birds. These findings partly do not agree with a previous study that demonstrated the presence of α2,3- and α2,6-linked sialic acids in quail and chicken intestinal cells [62]. It seems that both sialic acids are responsible for the binding of influenza A virus to human type receptors and, consequently, that the digestive tracts of chickens and quails are sites in which new influenza viruses with human pandemic potential may be generated [62]. Probiotic administration increased the number and the staining intensity of ileal SNA-positive GCs and, on the contrary, it decreased the staining intensity of caecal GCs. Sialidase-PNA revealed the presence of O-linked sialoglycans containing the disaccharide Galβ1,3GalNa in the GCs of all intestinal tracts. The administrated probiotics increased the number of GCs containing this sialomucin in the ileum and caecum, whereas its content (staining intensity of sPNA) decreased in the ileum. A higher secretion of intestinal sialomucins has been related to improved defense in mammals [63]. It has been found that sialic acid is preferentially removed by *Clostridium perfringes* when this opportunistic pathogen, responsible for necrotic enteritis in poultry, was incubated with the mucin O-linked glycans from the small intestine of *Gallus gallus domesticus* [64].

In this study neutral O-glycans were detected with PNA that binds core 1 (T-antigen) Galβ1,3GalNAc [65]. This terminal disaccharide was found in the GCs throughout the intestine of both the Cand P guinea fowls. The probiotic supplementation induced an increase of the number of PNA-positive GCs in duodenum and caecum as well as an increase in Galβ1,3GalNAc content (PNA staining intensity) in duodenum and ileum. Since galactose and GalNAc constituting O-linked mucus in the small intestine of *Gallus gallus domesticus* are a carbon source for bacteria [64], these results suggest that the used probiotics could increase the energy source for the intestinal microbiota of guineafowls.

Goblet cells containing highly mannosilated N-linked glycans (Con A reactivity) [66] were distributed along the entire intestine, although to a minor extent compared with the other investigated sugars. The presence of high mannose N-glycans agrees with the constant presence of intracellular glycoprotein synthesis machinery, such as rough endoplasmic reticulum (RER) and Golgi apparatus, described in the ultrastructural studies of turkey poults [67]. In the probiotic-fed samples the number of Con A-positive CGs increased particularly in the ileum and caecum, and the staining intensity significantly increased in all intestinal tracts. The presence of mannosylated glycans as a source of carbon for bacteria has been found in the mucus of *Gallus gallus domesticus* [64].

With regards to the fucosylated mucins, the GCs of guinea fowl intestine were lacking in αl,2-fucosylated mucins (UEA I negativity), whereas they contained the fucosylated glycans reacting with LTA. This result partly agrees with a liquid chromatography–mass spectrometry (LC–MS) study on the chicken intestine, which demonstrated the presence, except for the caecum, of glycans with 1,2-fucose and/or 1,3-fucose and 1,4-fucose linkages [46]. It has been found that L-fucose provides a favorable ecological niche for specific commensal organisms capable of utilizing fucose as a carbon source [68,69]. However, fucosylated glycans are also involved in host–microbe interactions in the gastrointestinal tract [70]. Probiotic administration increased significantly the number of LTA-positive GCs only in duodenum that also displayed a staining intensity increase, which, on the contrary, decreased in the ileum and the caecum. The reduction in LTA-FITC intensity could depend on the higher secretion of fucosylated mucins compared to the duodenum.

The cytoplasm of the villous epithelial (columnar) cells of all the investigated intestinal tracts from both C and P birds contained neutral N-linked glycans with αl,2-fucosylated chains (PAS, Con A and UEA I reactivity). Fucose is a monosaccharide found in various biological tissues including glycans in the intestinal epithelium [70]. Previous transmission electron microscopy observations in the villous epithelial cells of turkey poults demonstrated the presence of the rough endoplasmic reticulum (RER) and vesicular bodies [71]. Both are ultrastructural evidence of the glycoprotein synthesis and the digestive processes. N-linked glycans are normal constituent of cells involved in absorptive and digestive processes and Con A-binding sites in epithelial cells haves been associated to these functions [72].

The luminal surface of the villous epithelial cells also stained with AB2.5, when compared to the cytoplasm of the epithelial cells. This indicates that the N-glycans produced in the cytoplasm of the columnar cells are enriched with acidic carboxylated (sialylated) glycoproteins moving to luminal surface. The intestinal luminal surface is constituted of a brush border that consists of glycocalix coated microvilli involved in the cell’s absorptive processes. It has been demonstrated that the presence of sialoglycoconjugates, due to negative charge of sialic acids, are involved in binding and transport of positively charged molecules as well as in attraction and repulsion phenomena between cells and molecules [73].

The epithelial cells of intestinal crypts showed the presence of high-mannose N-linked glycans, αl,2-fucosylated glycans, and O-linked glycans terminating with Galβ1,3GalNAc (Con A, LTA and PNA reactivity, respectively) without significant difference between C and P. Interestingly, the number of PNA-positive cells was smaller than Con A- and LTA-positive ones, suggesting that the crypts are made up of several cell types. This agrees with the view that intestinal crypts include pluripotent stem cells and cells that differentiate into mature cell lineages during migration along the crypts, such as the enterocytes and the mucin secretory goblet cells. Moreover, the crypts also showed cells containing sulfated and non-sulfated acidic glycans (HID and AB2.5 staining). These cells were similar in appearance and distribution to PNA-positive cells, but their number increased in the P guinea fowls. Since HID and AB2.5 positivity was observed only in the GCs, this finding suggests that PNA-positive cells could be precursors of the GCs. The presence of sulfated and non-sulfated acidic glycanshas been reported in the intestine of free-range chickens, whose dietary insect meal inclusion did not influence intestinal mucin composition [42].

## 5. Conclusions

In conclusion, this study demonstrates that dietary multi-strain probiotic Slab51^®^ supplementation induces changes in the gut morphology and affects in a region-specific manner the composition and content of glycoprotein, in particular the goblet cells mucins, of guinea fowls intestine. Since probiotics affect the mucin dynamics via transcriptional regulation [1,41,74], these results together with the positive influence on microbiota composition observed in a recent study of ours [18] suggest a promising use of the examined commercial blend for health preservation and growth performance of guinea fowls.

## Figures and Tables

**Figure 1 animals-11-00495-f001:**
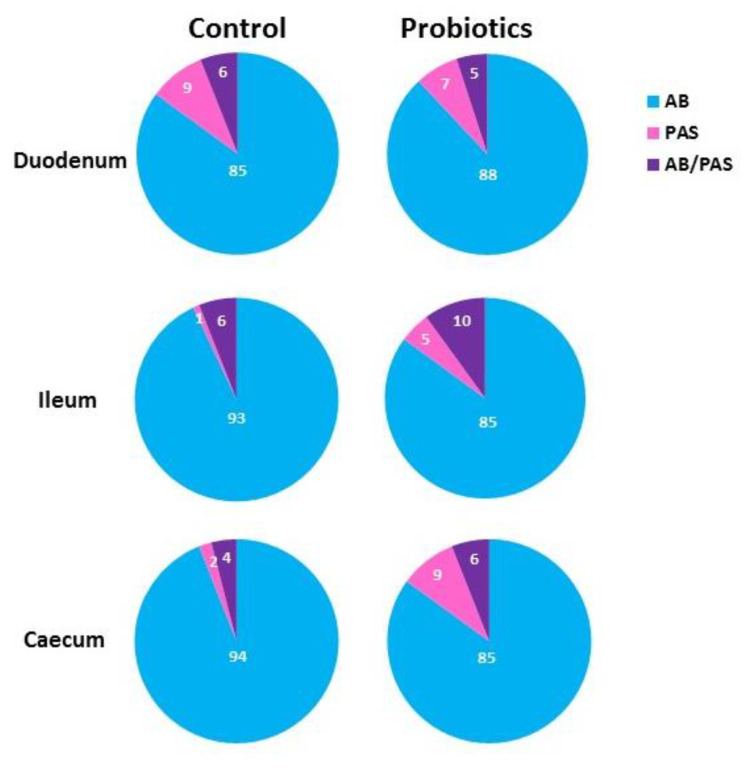
Percentage of the intestinal goblet cells of C and P guinea fowls stained with Alcian Blue pH2.5 (AB) (blue), PAS (magenta) and both AB2.5 and PAS staining (violet).

**Figure 2 animals-11-00495-f002:**
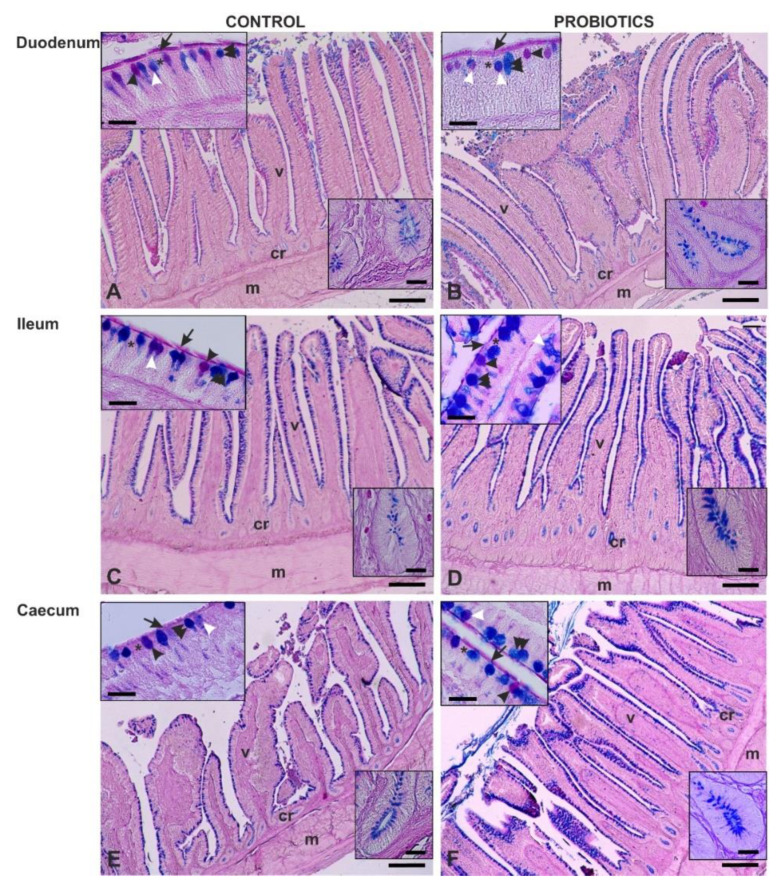
Alcian Blue pH 2.5 (AB2.5) and PAS staining of guinea fowl duodenum (**A**,**B**), ileum (**C**,**D**) and caecum (**E**,**F**). A,C,E: C samples; B,D,F: P samples. Most of the goblet cells exhibit AB 2.5 positivity (blue), whereas few goblet cells were either only PAS-positive (magenta) or both PAS- and AB 2.5-positive (violet). Insets show details of villus (top) and crypt (bottom) epithelial cells. cr, crypt; m, muscularis wall; v, villus; arrow, luminal surface; asterisk, epithelial cell. Black arrowhead, PAS-positive goblet cells; double arrowhead, AB2.5-positive goblet cells; white arrowhead, mixed stained goblet cells. Scale bars: A–F, 200 µm. Insets: A–F, 20 µm.

**Figure 3 animals-11-00495-f003:**
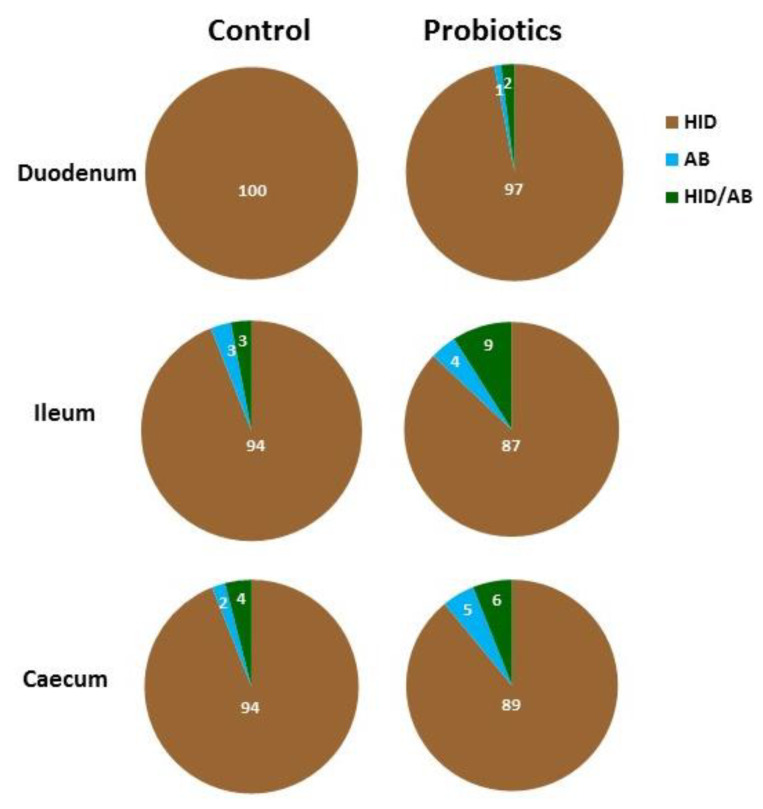
Percentage of goblet cells stained with HID (brown), Alcian Blue pH 2.5 (AB) (blue) and both HID and Alcian Blue pH 2.5 (green) in the villi of guinea fowl intestine.

**Figure 4 animals-11-00495-f004:**
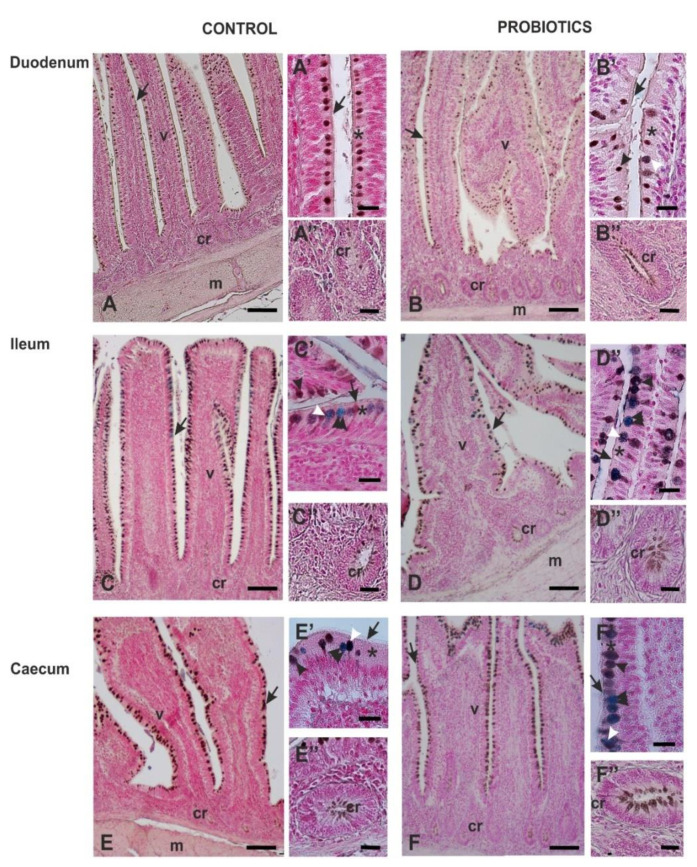
HID/AB2.5 staining sequence of guineafowl duodenum (**A**,**B**), ileum (**C**,**D**) and caecum (**E**,**F**). A,C,E: C samples; B,D,F: P samples. Most of the goblet cells exhibit HID positivity (brown), whereas few goblet cells are either only AB2.5-positive (blue) or both HID- and AB2.5-positive. cr, crypt; m, muscularis wall; v, villus; arrow, villus luminal surface; asterisk, epithelial cells. Black arrowhead, HID-positive goblet cells; double arrowhead, AB2.5-positive goblet cells; white arrowhead, mixed stained goblet cells. Scale bars: A–F, 100 µm. A’–F’ and A’’–F’’, 20 µm.

**Figure 5 animals-11-00495-f005:**
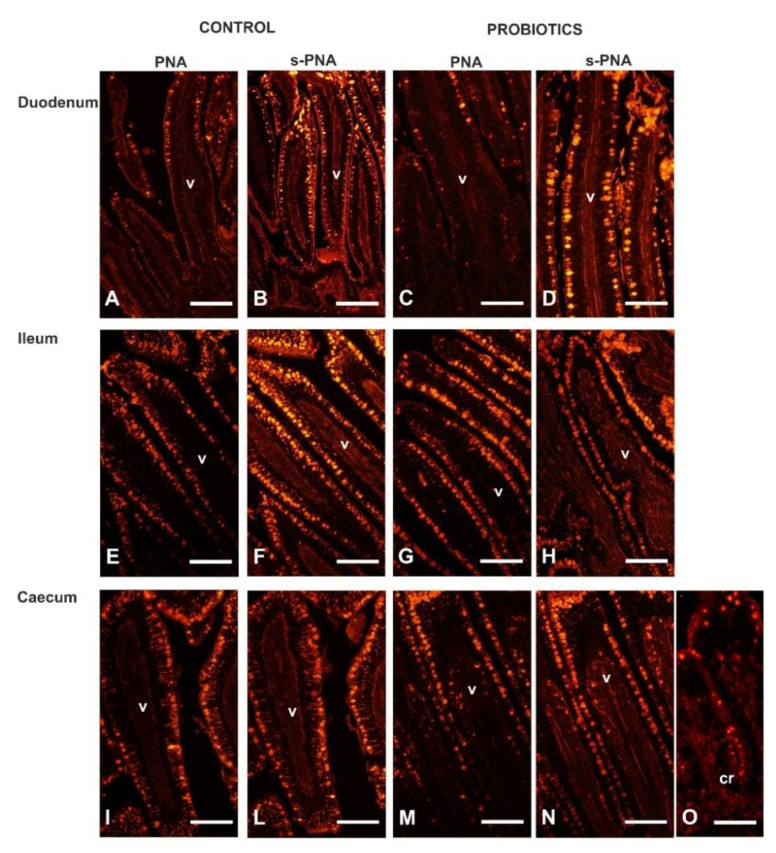
Reactivity pattern of Peanut agglutonin (PNA) (**A**,**C**,**E**,**G**,**I**,**M**,**O**) and sialidase (s)PNA (**B**,**D**,**F**,**H**,**L**,**N**) with intestinal mucosa of Cand P guinea fowls. cr, crypt; v, villus. Scale bars: A–O, 100 µm.

**Figure 6 animals-11-00495-f006:**
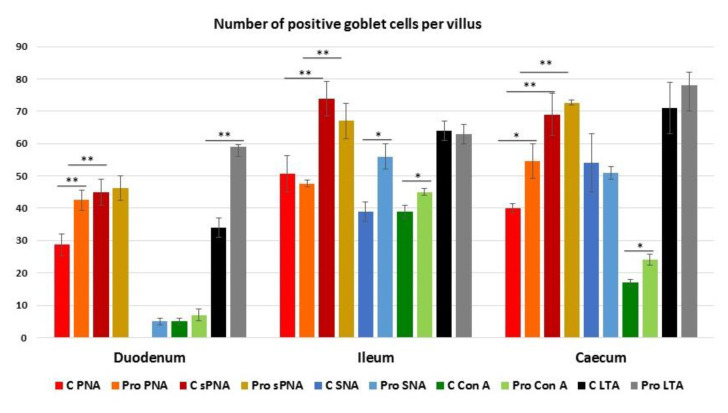
Number of goblet cells of C and P in guinea fowl intestine stained with Peanut agglutinin (PNA), sialidase (s)-PNA, Sambucus nigra agglutinin (SNA), Concanavalin A (Con A) and Lotus tetragonolobus agglutinin (LTA). Data show the mean with error bars representing ± SE and Student’s *t*-test results: * *p* < 0.05; ** *p* < 0.001.

**Figure 7 animals-11-00495-f007:**
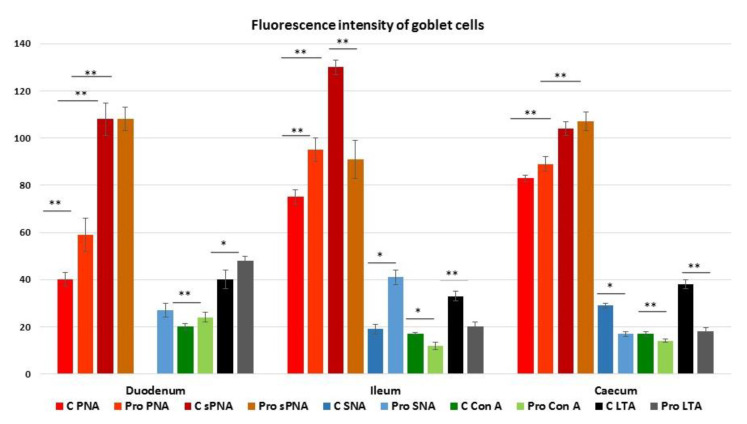
Quantification of Peanut agglutinin (PNA), sialidase (s)PNA, Sambucus nigra agglutinin (SNA), Concanavalin A (Con A) and Lotus tetragonolobus (LTA) fluorescence signals in the intestinal goblet cells of C and P guinea fowls. Data show the mean with error bars representing ± SE and Student’s *t*-test results: * *p* < 0.05; ** *p* < 0.001.

**Figure 8 animals-11-00495-f008:**
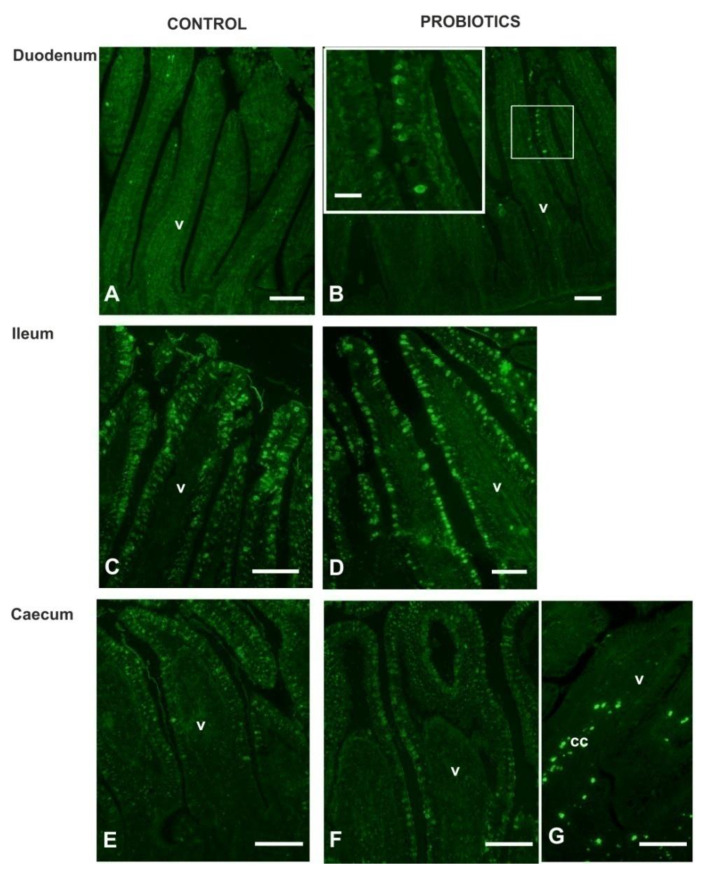
Reactivity pattern of SNA (**A**–**F**) and MAL II (**G**) with intestinal mucosa of guinea fowls. Inset of B shows magnification of the square-marked zone. cc, connective tissue cells. Scale bars: A–G, 100 µm.

**Figure 9 animals-11-00495-f009:**
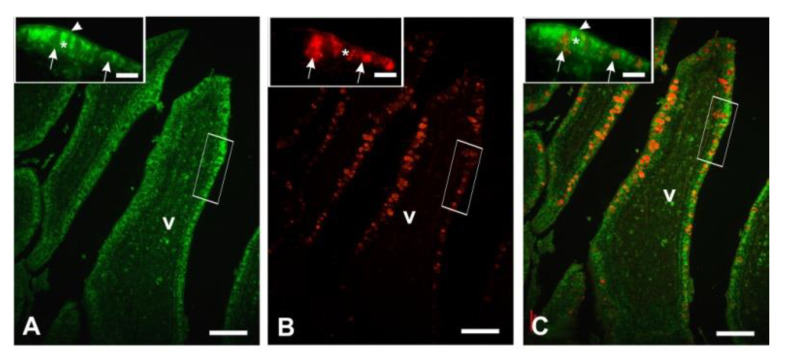
Con A-FITC (**A**), PNA-TRITC (**B**), Con A-FITC and PNA-TRITC merged pictures (**C**) showing the faintly visible Con A-positive GCs in the duodenum of guinea fowls. Insets represent the high magnification of rectangle-marked zones. v, villus; arrow, goblet cell; arrowhead, luminal surface; asterisk, epithelial cell. Scale bars: A,B,C, 100 µm; insets: 40 µm.

**Figure 10 animals-11-00495-f010:**
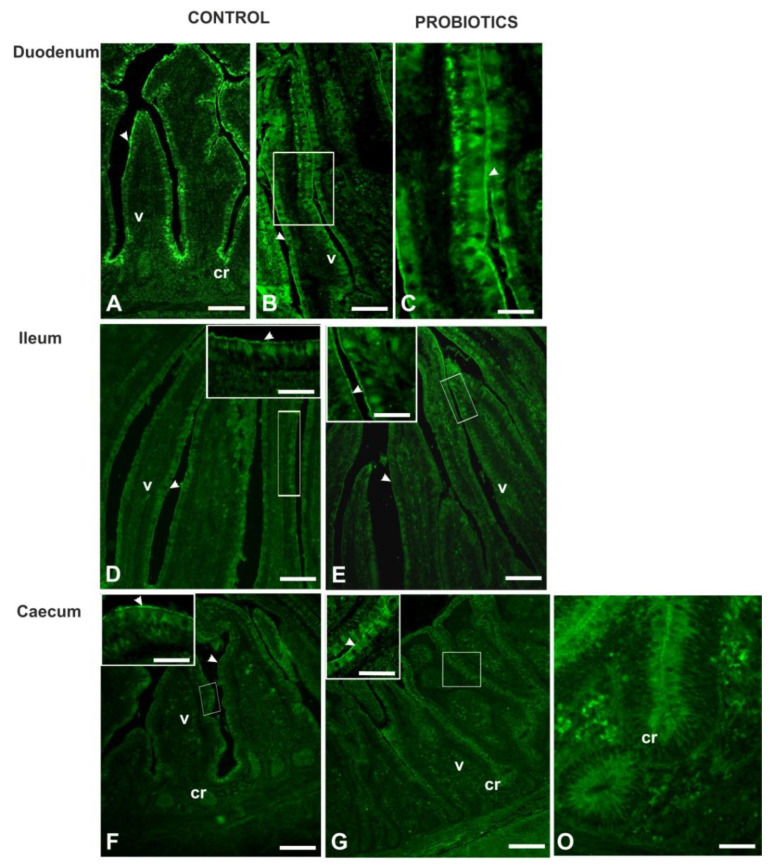
Representative reactivity of Con A with intestinal mucosa of C (**A**,**D**,**F**) and P (**B**,**C**,**E**,**G**,**O**) guinea fowls. C shows high magnification of the square-marked zone from B. Insets in D,E,F,G show detail of marked zones. cr, crypt; v, villus; arrowhead, luminal surface. Scale bars: A,D,E,F,G, 80 µm; B, 100 µm; C, 200 µm; insets D,E,F, 180 µm; O and inset G, 150 µm.

**Figure 11 animals-11-00495-f011:**
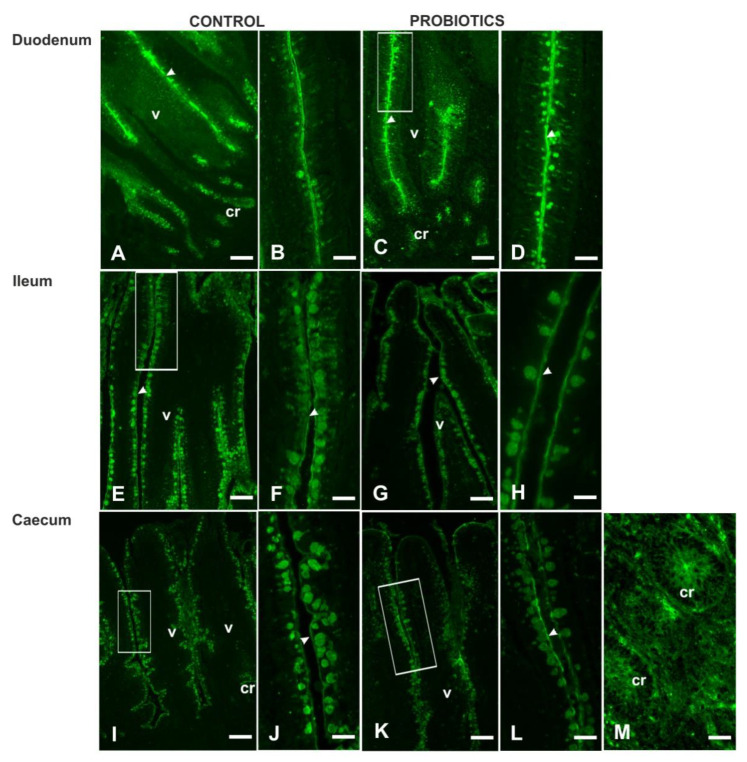
Fucosylated glycans in the intestine of C (**A**,**B**,**E**,**F**,**I**,**J**) and P (**C**,**D**,**G**,**H**,**K**,**L**,**M**) guinea fowls showed with LTA. D,F,J,L display high magnification of the rectangle-marked zone from C,E,I,K. cr, crypt; v, villus; arrowhead, luminal surface. Scale bars: A,C,I,K, 100 µm; B,D,J,L, 30 µm; E, 50 µm; F,H, 25 µm; G, 70 µm; M, 120 µm.

**Table 1 animals-11-00495-t001:** Chemical composition of the two feeds used during the grow-out cycle in the two groups (C, P) of guinea fowls.

Chemical Composition (%)	I Period (10–40 Day)	II Period (41–120 Day)
Protein	22	21
Lipids	5	5.8
Ash	7.4	7.5
Cellulose	4.4	4.5
Ca	1	1.04
P	1	0.85
Na	0.2	0.19
Lysine	1.2	1.2
Methionine	0.6	0.58
Specific phytase promoter	1500	750
Digestion promoterEndo-1.4-ß-Xylanasi	200	0
Vitamin A E672 (UI/kg)	0	4000
Vitamin D3 E671 (UI/kg)	2000	1250
Vitamin E (alpha toc. 91%) (mg/kg)	40	20
Cu E4 (mg/kg)	16	10
Se E8 (mg/kg)	0.16	0.2
Luteine E161b (g/kg)	0	41
Zeaxanthine E161 (g/kg)	0	8.4

**Table 2 animals-11-00495-t002:** Lectin used, their sugar specificities and the inhibitory sugars used in control experiments.

Lectin Abbreviation	Source of Lectin	µg/mL	Sugar Specificity	Inhibitory Sugar
MAL II	*Maackia amurensis*	25	NeuNAcα2-3Galβ1-3(± NeuNAcα2,6)GalNAc	NeuNAc
SNA	*Sambucus nigra*	25	Neu5Acα2,6Gal/GalNAc	NeuNAc
Con A	*Canavalia ensiformis*	25	Terminal/internal αMan > αGlc	Mannose
PNA *	*Arachis hypogaea*	25	Terminal Galβl,3GalNAc	Galactose
LTA	*Lotus tetragonolobus*	30	Terminal *a* L-Fuc	Fucose
UEA I	*Ulex europaeus*	30	Terminal L-Fucαl,2Galβl,4GlcNAcβ	Fucose

Fuc, fucose; Gal, galactose; GalNAc, N-acetylgalactosamine; Glc, glucose; GlcNAc, N-acetylglucosamine; Man, mannose; NeuNAc, N-acetylneuraminic (sialic) acid. * TRITC (rhodamine)-labeled lectin. Non-marked lectins were FITC (fluorescein isothiocyanate)-labeled lectins.

**Table 3 animals-11-00495-t003:** Effect ofthe multi-strain probiotic Slab51^®^ supplementationon the intestinal morphology of guinea fowl (*Numida meleagris*).

Item	Control	Probiotics	*p*-Value
**Duodenum**				
Villus height (µm)	649.11 ± 22.71	895.71 ± 16.46	<0.001
Crypt depth (µm)	101.58 ± 27.62	143.38 ± 8.72	<0.001
Goblet cell numbers per villus	122.92 ± 3.4	169.11 ± 6.4	<0.001
**Ileum**				
Villus height (µm)	671.88 ± 15.41	747.52 ± 20.18	0.010
Crypt depth (µm)	110.97 ± 5.31	126.58 ± 7.93	0.002
Goblet cell numbers per villus	139.24 ± 5.35	155.73 ± 1.46	<0.001
**Caecum**				
Villus height (µm)	464.81 ± 60.02	594.97 ± 68.31	0.043
Crypt depth (µm)	84.82 ± 6.81	119.10 ± 9.44	<0.001
Goblet cell numbers per villus	94.09 ± 2.12	141.15 ± 6.71	<0.001

Values are expressed as means ± SE. The number of goblet cells was determined by staining sections with AB 2.5/PAS sequence.

## Data Availability

The datasets used and analyzed in the current study are available from the corresponding authors on reasonable request.

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
