# Peer review of "Modulation of Morphology and Glycan Composition of Mucins in Farmed Guinea Fowl (*Numida meleagris*) Intestine by the Multi-Strain Probiotic Slab51^®^"

_animals, 2021, doi:10.3390/ani11020495_

Round 1
Reviewer 1 Report
This manuscript describes a detailed morphological analysis of intestinal samples from Guinea fowl birds fed either a standard diet (Control) or probiotic-supplemented diet (Probiotic). While the scientific approach is valid, the presentation detracted from its appeal mostly due to the lack of consistency and weak English. It’s beyond the review scope to list few pages of technical edits, therefore I highly recommend having a strong and detailed revision of the English for typos, terminology, verbiage, sentence structure, punctuation, spaces, and consistency throughout the manuscript.
Specific comments
Simple Summary
L16: ‘In the poultry…’
L24: these birds are guineafowl not Guinea fowl (i.e. one word in scientific publication) and Numida meleagris are specifically the helmeted guineafowl. All occurrences must be corrected.
L24: the unit (g) should follow the weight and not precede it.
L24-25: what is a ‘normal livestock cycle’ in this context? Do you mean a grow-out cycle? Typically for guineafowl, this is around 80 days of age (or 90-95 d for free range birds), so why raise them > 1 month longer?! Is this truly relevant to real field application?
Abstract
L32: here is an example of word selection ‘…considered synonymous with animal health…’
L33: how long were the birds fed the probiotic? What was the inclusion rate?
L33-34: did each group have 20 birds? Were all 40 birds weighed individually and sampled?
L36: a brief statistics statement needs to be included.
L37: ‘villus depth of crypts’ should simply be ‘crypt depths’
L38-41: all abbreviations need to either spelled out or the defined (here and upon first occurrence in the text).
L42-43: ‘increase of crypts cells’???
Introduction
L53: replace ‘flora’ with microbiota
L85: ‘immune system’ not ‘immune-system’
L92: these are not ‘a new promising species’ as they’ve been raised for this purpose for decades!
Materials & Methods
L104: relocate the unit (g) here and throughout.
L104-105: please define the groups here and use consistently throughout the manuscript including figures/tables (i.e. Control and Probiotic), there are several terms used which gets confusing.
L114: ‘with litter on the bottom’ is not appropriate, need to specific what type of litter was used, how much was used (i.e. thickness), fresh or pre-used, etc.
L114-115: specify the ‘controlled photoperiod’ (i.e. number of light/dark hours)
L115: did you mean ‘natural ventilation?’ were these open sheds, windows, curtain, etc.
L116-117: ‘the same commercial pellet feed’? Chicks cannot eat pelleted feed for first few days, typically they are fed crumbled feed. What was the composition of this feed? Commercially, birds are fed starter, grower (sometimes 1 and 2), and finisher diets. All such details need to be provided, the proximate analysis (or at least formulated diets) are best presented in a table.
L120-121: again, were all 40 birds weighed (not weighted) and sampled individually?
L124: which areas of each intestinal sections were collected? i.e. the 3-cm pieces.
L141: need to spell out RT (assuming it is room temperature)
L173: what does ‘to render the enzyme…’ mean? To activate it?
L189- ‘non-stained’
Results
L198: not sure what ‘satisfactory’ mean! Normal growth/body weights? Were the birds weighed and recorded individually or a group of 20? Were the initial weights (day 1) taken to calculate weight gains? Was the feed consumption measured?
L203: per earlier comment above (L104-105), change No-Pro to Control and Pro to Probiotic. In other places C and P are used!
L209-221: the data presented in figures 1 & 2 are better presented in a table format with clear statistics (P values and SEM).
L213: the ‘t-’ in ‘t-test’ should be italic.
L287-288: what does ‘the latter intestinal tract’ mean? If that’s the caeca, then it needs to be spelled out.
L293: change ‘samples’ to ‘birds’ as one doesn’t feed samples.
L362: in which group were the GCs more abundant?
L389: what does ‘induced the appearance’ mean? Expression, staining, etc.?
L396: ‘did not react’
L433: Con A was not defined anywhere earlier; all such abbreviations need to be defined upon first occurrence.
Discussion
Generally, this section is too long relative to the data presented herein, and this is largely due to redundancy and repetition of the results. The Discussion can be easily reduced by third or even half without losing any relevant content, therefore it needs extensive revision.
L483-484: revise this statement. As commented above, use consistent terminology and explain what’s meant by ‘satisfactory.’
L484-485: ‘…, slightly higher respect to C’ makes little sense, perhaps you meant to say ‘which were slightly higher compared to Control’ or ‘…slightly higher than those of Control birds’
L485: ‘although not statistically significant’ instead of ‘although not significant from the statistical point of view.’
L490: not sure ‘zootechnical’ is the appropriate term here!
L515: ‘activity of GCs’…there is a significant number of such phrasing throughout that needs to be addressed.
L572-576: this is an example of an irrelevant statement that can be deleted.
L589: what do the authors mean by ‘native’?
L594-602: another irrelevant paragraph.
L701: until it is accepted, this reference should not be cited.
References
Consistency here is key, please follow the journal guidelines.
Author Response
REVIEWER 1
Answer for the reviewer’s comments:
Dear Reviewer,
Thank you for all the comments and suggestions which helped us to improve the quality of our manuscript. All the corrections were included into the text of the manuscript and are shown in the "Track Changes" .
This manuscript describes a detailed morphological analysis of intestinal samples from Guinea fowl birds fed either a standard diet (Control) or probiotic-supplemented diet (Probiotic). While the scientific approach is valid, the presentation detracted from its appeal mostly due to the lack of consistency and weak English. It’s beyond the review scope to list few pages of technical edits, therefore I highly recommend having a strong and detailed revision of the English for typos, terminology, verbiage, sentence structure, punctuation, spaces, and consistency throughout the manuscript.
Specific comments
Simple Summary
L16: ‘In the poultry…’
Au: corrected.
L24: these birds are guineafowl not Guinea fowl (i.e. one word in scientific publication) and Numidameleagris are specifically the helmeted guineafowl. All occurrences must be corrected.
Au:We used the term “guinea fowl”and we corrected the term in all the text as in the grow-out cycle was used thedomesticated strain selected in Europe, and not specifically the wild-type bird (helmeted guinea folw). The choice to use 2 words was taken according to scientific papers published in Poultry Science (Yamak et al., 2018; Tufarelli et al., 2015; Laudadio et al., 2012).
L24: the unit (g) should follow the weight and not precede it.
Au: done.
L24-25: what is a ‘normal livestock cycle’ in this context? Do you mean a grow-out cycle? Typically for guineafowl, this is around 80 days of age (or 90-95 d for free range birds), so why raise them > 1 month longer?! Is this truly relevant to real field application?
Au: the term “grow-out cycle” has been inserted. The growing cycle lasted 120 days because this was the target to reach a mean body weight of at least 1700 g.
Abstract
L32: here is an example of word selection ‘…considered synonymous with animal health…’
Au: done.
L33: how long were the birds fed the probiotic? What was the inclusion rate?
Au: The birds received the probiotics in drinking water (2x1011 UFC/L) during all the grow-out cycle. We modified the abstract, accordingly.
L33-34: did each group have 20 birds? Were all 40 birds weighed individually and sampled?
Au: yes, the individual weight was relieved.
L36: a brief statistics statement needs to be included.
Au: statistics statement was added.
L37: ‘villus depth of crypts’ should simply be ‘crypt depths’
Au: corrected.
L38-41: all abbreviations need to either spelled out or the defined (here and upon first occurrence in the text).
Au: abbreviations were deleted.
L42-43: ‘increase of crypts cells’???
Au: we corrected to “crypt cells”.
Introduction
L53: replace ‘flora’ with microbiota
Au: done.
L85: ‘immune system’ not ‘immune-system’
Au: done.
L92: these are not ‘a new promising species’ as they’ve been raised for this purpose for decades!
Au: the sentences was replaced with “a suitable alternative to hens housed in free range system”.
Materials & Methods
L104: relocate the unit (g) here and throughout.
Au: done.
L104-105: please define the groups here and use consistently throughout the manuscript including figures/tables (i.e. Control and Probiotic), there are several terms used which gets confusing.
Au: we define all the groups C and P throughout the manuscript.
L114: ‘with litter on the bottom’ is not appropriate, need to specific what type of litter was used, how much was used (i.e. thickness), fresh or pre-used, etc.
Au: done.
L114-115: specify the ‘controlled photoperiod’ (i.e. number of light/dark hours)
Au: done.
L115: did you mean ‘natural ventilation?’ were these open sheds, windows, curtain, etc.
Au: done.
L116-117: ‘the same commercial pellet feed’? Chicks cannot eat pelleted feed for first few days, typically they are fed crumbled feed. What was the composition of this feed? Commercially, birds are fed starter, grower (sometimes 1 and 2), and finisher diets. All such details need to be provided, the proximate analysis (or at least formulated diets) are best presented in a table.
Au: the table dedicated to compositionof the two feeds has been introduced in the manuscript (Table 1).
L120-121: again, were all 40 birds weighed (not weighted) and sampled individually?
Au: the “individually” adverb has been added.
L124: which areas of each intestinal sections were collected? i.e. the 3-cm pieces.
Au: the specific areas have been added.
L141: need to spell out RT (assuming it is room temperature)
Au: done.
L173: what does ‘to render the enzyme…’ mean? To activate it?
Au: Sialic acid could have N- and O-acetylated groups. O-acetylation of sialic acid makes it sialidase-insensitive. Therefore, we use the sentence “to render the enzyme digestion effective” because the saponification increases the cleavage of most of the sialic acids, including O-acetylated sialic acids.
L189- ‘non-stained’
Au: done.
Results
L198: not sure what ‘satisfactory’ mean! Normal growth/body weights? Were the birds weighed and recorded individually or a group of 20? Were the initial weights (day 1) taken to calculate weight gains? Was the feed consumption measured?
Au: “satisfactory” has been changed with “favourable”. In M&M, we have just been added the individual body weight was recorded. Unfortunately, the initial weights are not available because the farm housed the birds when they were 10 days old. In this way, we are not able to report the feed conversion rate.
L203: per earlier comment above (L104-105), change No-Pro to Control and Pro to Probiotic. In other places C and P are used!
Au: we used now C and P in all the manuscript.
L209-221: the data presented in figures 1 & 2 are better presented in a table format with clear statistics (P values and SEM).
Au: the data are presented in the Table 3.
L213: the ‘t-’ in ‘t-test’ should be italic.
Au: done.
L287-288: what does ‘the latter intestinal tract’ mean? If that’s the caeca, then it needs to be spelled out.
Au: ‘the latter intestinal tract’ was changed to “The caecum”.
L293: change ‘samples’ to ‘birds’ as one doesn’t feed samples.
Au: done.
L362: in which group were the GCs more abundant?
Au: we wrote "C fowls".
L389: what does ‘induced the appearance’ mean? Expression, staining, etc.?
Au: "Appearance" was replaced with "staining".
L396: ‘did not react’
Au: done.
L433: Con A was not defined anywhere earlier; all such abbreviations need to be defined upon first occurrence.
Au: revised as requested.
Discussion
Generally, this section is too long relative to the data presented herein, and this is largely due to redundancy and repetition of the results. The Discussion can be easily reduced by third or even half without losing any relevant content, therefore it needs extensive revision.
Au: the Discussion was revised and reduced.
L483-484: revise this statement. As commented above, use consistent terminology and explain what’s meant by ‘satisfactory.’
Au: done.
L484-485: ‘…, slightly higher respect to C’ makes little sense, perhaps you meant to say ‘which were slightly higher compared to Control’ or ‘…slightly higher than those of Control birds’
Au: done.
L485: ‘although not statistically significant’ instead of ‘although not significant from the statistical point of view.’
Au: done.
L490: not sure ‘zootechnical’ is the appropriate term here!
Au: “Zootechnical” has been changed with “productive”.
L515: ‘activity of GCs’…there is a significant number of such phrasing throughout that needs to be addressed.
Au: we corrected according to the suggestion.
L572-576: this is an example of an irrelevant statement that can be deleted.
Au: the sentence was deleted.
L589: what do the authors mean by ‘native’?
Au: “native” was deleted.
L594-602: another irrelevant paragraph.
Au: Lines 594-598 “Eukaryotic glycans have been classified into two major classes: N-glycans and O-glycans. The carbohydrate side chains of the protein backbone of intestinal mucins are composed of N-acetilneuraminic acid (NeuNAc, sialic acid), N-acetylgalactosamine (GalNAc), N-acetylglucosamine (GlcNAc), Galactose (Gal), Glucose (Glc), and Mannose (Man), and Fucose (Fuc) [2,3].” were deleted. The remaining text has been retained to explain why the lectins were used.
L701: until it is accepted, this reference should not be cited.
Au: the article was deleted.
References
Consistency here is key, please follow the journal guidelines.
Au: the references were corrected according to the guidelines.
Reviewer 2 Report
This manuscript described the effects of dietary multistrain probioticSlab51® on the morphology and mucin glycan composition of glycoproteins produced by goblet cells in the duodenum, ileum and cecum of Guinea fowl using histological techniques. The results showed the probiotics treatment improved intestinal mucosal structure and modified composition of glycoproteins of mucin. Specifically, probiotics increased the number of GCs containing sialomucin as well as increase in PNA and ConA positive GCs number and staining intensity. These data suggests that the probioticSlab51® may enhance intestinal health of guinea fowl.
The manuscript is well written and experiment is elaborated. The results including significant and interesting information for poultry industry, however the manuscript is including some issues should be addressed, as below.
Major
- The results clearly showed that the probiotics changed composition of the glycoprotein. What is the mechanism how the probiotic bacteria changed composition of glycoprotein produced from GC?
- L291-293 and 349-351, Authors described that GC in crypt showed stronger stained and larger number of positive cells in probiotics group than in the control group. Did you do any histological and statistical analysis of this point?
Minor
L103, Did you check sex of the experimental birds when they were slaughtered?
L114,115, Authors should add concrete light and dark periods.
L188-189, How many sections and fields of view were analyzed for each sample?
Figure 4 and 6, Please indicate the different staining pattern of GC cells (single and double positive cells) by different types of arrows.
Author Response
REVIEWER 2
Dear Reviewer,
Thank you for all the comments and suggestions which helped us to improve the quality of our manuscript. All the corrections were included into the text of the manuscript and are shown in the "Track Changes" .
This manuscript described the effects of dietary multistrain probioticSlab51® on the morphology and mucin glycan composition of glycoproteins produced by goblet cells in the duodenum, ileum and cecum of Guinea fowl using histological techniques. The results showed the probiotics treatment improved intestinal mucosal structure and modified composition of glycoproteins of mucin. Specifically, probiotics increased the number of GCs containing sialomucin as well as increase in PNA and ConA positive GCs number and staining intensity. These data suggests that the probioticSlab51® may enhance intestinal health of guinea fowl.
The manuscript is well written and experiment is elaborated. The results including significant and interesting information for poultry industry, however the manuscript is including some issues should be addressed, as below.
Major
The results clearly showed that the probiotics changed composition of the glycoprotein. What is the mechanism how the probiotic bacteria changed composition of glycoprotein produced from GC?
Au: Probiotics affect the mucin dynamics via transcriptional regulation (Kim and Ho, 2010; Smirnov et al., 2005; Aliakbarpour et al., 2012). This sentence was added in the Conclusions.
L291-293 and 349-351, Authors described that GC in crypt showed stronger stained and larger number of positive cells in probiotics group than in the control group. Did you do any histological and statistical analysis of this point?
Au: No in-depth analysis was made in this regard because the villi were the target of our investigation.
Minor
L103, Did you check sex of the experimental birds when they were slaughtered?
Au: due to a very fast slaughtering process, the sex was not determined.
L114,115, Authors should add concrete light and dark periods.
Au: photoperiod was added.
L188-189, How many sections and fields of view were analyzed for each sample?
Au: all fields of at least 3 sections for each sample and staining were analyzed.
Figure 4 and 6, Please indicate the different staining pattern of GC cells (single and double positive cells) by different types of arrows.
Au: done.
Reviewer 3 Report
Dear Authors,
You mention in your introduction the role of gut health as a driver for a better animal health and welfare. You also indicate the relevance of mucosa of small intestine and necrotic enteritis and the interaction with several factors . One is the feed ingredient composition ( dietary compounds) .My request is to ask you if you could introduce the feed composition and also if that feed preparation has coccidiostats (comercial pellet feed from Cruciani Italy). I am saying that because some coccidiostats has a potential effect on clostridium perfringes.
Author Response
REVIEWER 3
Dear Reviewer,
Thank you for the suggestions which helped us to improve the quality of our manuscript. The corrections were included into the text of the manuscript and are shown in the "Track Changes" .
Dear Authors,
You mention in your introduction the role of gut health as a driver for a better animal health and welfare. You also indicate the relevance of mucosa of small intestine and necrotic enteritis and the interaction with several factors. One is the feed ingredient composition (dietary compounds). My request is to ask you if you could introduce the feed composition and also if that feed preparation has coccidiostats (comercial pellet feed from Cruciani Italy). I am saying that because some coccidiostats has a potential effect on clostridium perfringes.
Au: the feed composition has been added as new Table 1. Coccidiostats was not included in the pellet feed used in the grow-out cycle.
Round 2
Reviewer 1 Report
The authors did a nice job addressing the scientific issues but the English still needs work.
L39: The authors did not include a statistical statement in the abstract as claimed, only P values. A brief (one sentence) statement about how the data were analyzed should be added.
L47-48: do you mean increase in the numbers of crypt cells containing...?
L123: delete 'on the bottom'
L124: replace with 'Throughout the trial'
Author Response
Dear Reviewer,
Thank you for all the comments and suggestions which helped us to improve the quality of our manuscript. All the corrections were included into the text of the manuscript and are shown in the "Track Changes"
Comments and Suggestions for Authors:
The authors did a nice job addressing the scientific issues but the English still needs work.
Au: The English was improved.
L39: The authors did not include a statistical statement in the abstract as claimed, only P values. A brief (one sentence) statement about how the data were nalyzed should be added.
Au: The statement "The results were evaluated for statistical significance by Student’s t test." was added
L47-48: do you mean increase in the numbers of crypt cells containing...?
Au: the text was corrected according to the reviewer's suggestion
L123: delete 'on the bottom'
Au: done.
L124: replace with 'Throughout the trial'
Au: done.